# Combining Isotope Dilution and Standard Addition—Elemental Analysis in Complex Samples

**DOI:** 10.3390/molecules26092649

**Published:** 2021-04-30

**Authors:** Christine Brauckmann, Axel Pramann, Olaf Rienitz, Alexander Schulze, Pranee Phukphatthanachai, Jochen Vogl

**Affiliations:** 1Physikalisch-Technische Bundesanstalt (PTB), Bundesallee 100, 38116 Braunschweig, Germany; christine.brauckmann@ptb.de (C.B.); olaf.rienitz@ptb.de (O.R.); alexander.schulze@ptb.de (A.S.); 2Bundesanstalt für Materialforschung und-Prüfung (BAM), Richard-Willstätter-Str. 11, 12489 Berlin, Germany; pranee@nimt.or.th (P.P.); jochen.vogl@bam.de (J.V.); 3National Institute of Metrology (Thailand) (NIMT), 3/4-5 Moo 3, Klong 5, Klong Luang, Pathumthani 12120, Thailand

**Keywords:** isotope dilution mass spectrometry, standard addition, ICP–MS, blank characterization, silicon, sulfur, transferrin, tetramethylammonium hydroxide, biodiesel fuel, human serum

## Abstract

A new method combining isotope dilution mass spectrometry (IDMS) and standard addition has been developed to determine the mass fractions *w* of different elements in complex matrices: (a) silicon in aqueous tetramethylammonium hydroxide (TMAH), (b) sulfur in biodiesel fuel, and (c) iron bound to transferrin in human serum. All measurements were carried out using inductively coupled plasma mass spectrometry (ICP–MS). The method requires the gravimetric preparation of several blends (b*_i_*)—each consisting of roughly the same masses (*m*_x,*i*_) of the sample solution (x) and *m*_y,*i*_ of a spike solution (y) plus different masses (*m*_z,*i*_) of a reference solution (z). Only these masses and the isotope ratios (*R*_b,*i*_) in the blends and reference and spike solutions have to be measured. The derivation of the underlying equations based on linear regression is presented and compared to a related concept reported by Pagliano and Meija. The uncertainties achievable, e.g., in the case of the Si blank in extremely pure TMAH of *u*_rel_ (*w*(Si)) = 90% (linear regression method, this work) and *u*_rel_ (*w*(Si)) = 150% (the method reported by Pagliano and Meija) seem to suggest better applicability of the new method in practical use due to the higher robustness of regression analysis.

## 1. Introduction

Combined developments in inductively coupled plasma mass spectrometry (ICP–MS) techniques and instrumentation, with the continuous improvement of the calibration and evaluation of ICP–MS measurements, are currently considered the most accurate and precise tools in metrology in chemistry. This combination enables a remarkable decrease in measurement uncertainties in elemental analysis [1,2]. Related areas of analytical chemistry are geochemistry, environmental sciences, clinical chemistry, and forensics as well as fundamental metrology, to mention just a few [2,3,4,5,6,7]. Depending on sample composition (e.g., number of isotopes of the element of interest), concentration range, availability, experience of the analyst, availability of certified reference materials (CRMs), intended uncertainty, and other reasons, the determination of element content (in the following: mass fraction *w*) of a sample (x) can be conducted by a variety of calibration strategies [2]. Beneath simple external one-point calibration options with external standards [8], additional sophisticated methods have been developed. Among them, isotope dilution mass spectrometry (IDMS) concepts are usually the methods of choice, provided that the analyte element consists of at least two stable isotopes. The IDMS approach is considered a primary method in metrology in chemistry: it is fully understood and operated on the highest technological level. Moreover, it yields the smallest measurement uncertainties possible (most accurate and precise) according to the *Guide to the Expression of Uncertainty in Measurement* (GUM) [9]. Additionally, loss of the analyte after addition and equilibration of the spike does not affect the results. This method, based on the measurement of isotope ratios (intensity ratios), has been successfully applied in analytical chemistry for several decades and is still subject to further improvement [2,10,11,12,13]. Several variations of the IDMS method are known, based on the classical “single” IDMS approach. This needs a fully characterized spike material (same element as the analyte, with preferably inverted isotopic composition and known molar mass and purity) for the preparation of a blend consisting of the sample and the spike. However, often, the characterization of the spike is difficult or impossible, leading to “double” and higher-order (“triple”) IDMS approaches [2,14,15]. In “double” IDMS, a second blend is prepared from the spike and a well-characterized reference material, rendering the knowledge of purity and the molar mass of the spike obsolete. In “triple” IDMS, three blends are applied, with the additional advantage that the isotope ratio in the initial spike does not need to be measured. These higher-order approaches yield lower uncertainties at the cost of considerably more effort in sample preparation.

In the case of a monoisotopic system, the standard addition method is usually applied as an internal calibration method, being more sensitive and accurate than any external method [16,17,18]. This approach is chosen when the analyte is present in a complex matrix and/or a low concentration range. The regression analysis of the linear curve of the measured signal in several aliquots versus the ratio of the masses of the standard and sample aliquot yields in the *x*-axis intercept the desired mass fraction of the analyte element. A further-developed method is the standard addition method with an internal standard, useful in the case of long-term drifts and matrix effects [8,16]. This method requires the absence of the element of the internal standard in the sample, blank, and calibration standard. In most cases, the sample exists in a more or less different—and sometimes complex—matrix compared to the spike material.

Recently, Pagliano and Meija pioneered this problem in a sophisticated theoretical way by merging isotope dilution and standard addition methods [1]. They aimed at reducing or even circumventing matrix effects due to the matched matrix in all the measured solutions.

In this study, a similar and related method combining isotope dilution and standard addition methods has been developed independently. The intention of this work is to (1) reduce the impact of matrix effects in isotope dilution by combining it with a robust standard addition procedure, (2) simplify and linearize the working equations for the measurements of analyte mass fractions via linear regression to enable an uncertainty analysis according to GUM, (3) enable the determination of even trace amounts in, e.g., blank solutions, and (4) provide isotope ratio measurements with no need to correct for mass bias, which is a real benefit to the practical analyst. The method presented in the current work is demonstrated and validated experimentally using three completely different matrices in elemental analysis using ICP–MS: the determination of silicon with natural isotopic composition in an aqueous TMAH blank solution, the measurement of sulfur in biodiesel fuel (BDF), and the quantification of transferrin (TRF) in human serum.

A special need for the exact determination of the content of silicon with natural abundance—in aqueous TMAH used as a blank solution—is given in the *Avogadro Project* [19]. This was a multidisciplinary approach, initially set up for the realization of the Avogadro constant *N*_A_ with the lowest associated measurement uncertainty and, after the SI revision in 2019, aimed at the dissemination of the SI units of kilogram and mole [19,20,21,22]. In that context, chemically highly pure silicon, extensively enriched in ^28^Si, with *x*(^28^Si) > 0.9999 mol/mol, was used. For the determination of the respective molar mass *M* of the silicon, *u*_rel_(*M*) must be smaller than 1 × 10^−8^. This can only be achieved by reducing both the contamination and the quantification of the remaining natural silicon, which is described as an application of the new method presented.

The second example of the new combined method is the measurement of the mass fraction (*w*_x_) of sulfur in a biodiesel fuel matrix. These measurements were performed within an interlaboratory key comparison conducted by the IAWG of the CCQM (Inorganic Analysis Working Group of the Consultative Committee for Amount of Substance: Metrology in Chemistry and Biology): CCQM-K123 [23]. The mass fractions of several trace elements (impurities)—among them, sulfur—were determined. Originally, *w*_x_ was determined using an established IDMS technique. In addition, one of the biodiesel fuel samples was measured using the new combined IDMS-Standard Addition approach as a complementary method (this work). This enables the comparability of the new approach with an established and validated method. The biodiesel fuel matrix is an extremely complex matrix because of its volatility and impurities. Sulfur occurs in different compounds in biodiesel fuel, and, for obtaining accurate results, the conversion to sulfate must be ensured in the digestion step. Additionally, sulfur contamination during the whole analytical procedure is an issue and must be controlled.

In the third part, the quantification of TRF in human serum is presented as an example for speciation analysis with a clinical background. The origin of this work was a doctoral thesis followed by the EMRP HLT05 project, which was a European metrology research program [15]. TRF is a biomarker for congenital disorders of glycosylation as well as certain cancers and alcohol abuse, and, therefore, determining its concentration in blood aids in effective clinical diagnosis [24]. Due to this, TRF is listed as an important clinical analyte in the Guidelines of the German Medical Association of 2019 (Rili-BÄK) and is routinely measured in clinical laboratories [25] even though the target values are method-dependent since no reference method is available [25]. In clinical laboratories, the common way of quantifying TRF in serum is immune-based methods like immunoturbidimetry [26].

In order to quantify the protein TRF via its iron content, high-performance liquid chromatography (HPLC) was coupled to an ICP–MS. This approach enables us to separate TRF from the complex blood matrix and identify TRF by its characteristic retention time. To this day, the quantification of TRF is performed by applying double or triple IDMS and considering the iron background and complex matrix of the samples [15,27]. The new approach presented in this publication, combining IDMS with standard addition, is used for the first time in a complex biological sample as proof of principle. Since the sample is a certified serum material, the results of the new approach are compared with the certified target value and its reported range.

A central part of this work is also the comparison of the new method presented here with the one reported in [1]. Both methods have the same or a similar intention: the improvement of measurement design and the reduction of uncertainty for the determination of mass fractions *w*_x_ of elements in complex matrices. For better understanding and adaption to our notation, we have additionally derived the method reported in [1] from scratch for the three blends case: we ended up with exactly the same equation (which is Equation (9) in [1]). For the comparison, we evaluated all measurements using our new method (this work) and the one reported in [1].

## 2. Theoretical Methods

The main advantage of the combination of IDMS and standard addition is its unique simplicity of using gravimetrically prepared blends, ensuring traceability to the International System of Units (SI). It is based on the measurement of virtually only one isotope ratio and three masses per blend. The new approach yields, finally, Equation (1) to describe linear regression. The regression parameters, slope *a*_1_ and *y*-intercept *a*_0_, are then used to calculate the aimed-at mass fraction *w*_x_ of the analyte element in the original sample (x). The detailed derivation of Equations (1) and (2) is given in Appendix B.
(1)$$\underset{=y_{i}}{\underset{︸}{\frac{m_{y,i}}{m_{x,i}}\times \frac{R_{y}-R_{b,i}}{R_{b,i}-R_{x}}}}=\underset{=a_{1}}{\underset{︸}{\frac{1}{w_{y}}\frac{M_{y}}{M_{x}}\frac{\sum R_{y}}{\sum R_{x}}\times w_{z}}}\times \underset{=x_{i}}{\underset{︸}{\frac{m_{z,i}}{m_{x,i}}}}+\underset{=a_{0}}{\underset{︸}{\frac{1}{w_{y}}\frac{M_{y}}{M_{x}}\frac{\sum R_{y}}{\sum R_{x}}\times w_{x}}}$$
(2)$$w_{x}=\frac{a_{0}}{a_{1}}\times w_{z}$$
where *w*_x_ is directly proportional to the mass fraction *w*_z_ of a reference material (z; same element as the sample with the same isotopic composition as the analyte or, at least, very close to it); *m*_y,*i*_, *m*_x,*i*_, and *m*_z,*i*_ are the masses of a spike solution (y; same element as x with a preferably inverse isotopic composition), sample x, and reference material solution z, respectively. A number of blends (b*_i_*; e.g., five) containing almost the same masses of x and y, respectively, and different masses of z (*m*_z,1_ < *m*_z,2_ < *m*_z,3_ < *m*_z,4_ < *m*_z,5_) are required for the experiment and evaluation (Figure 1 and Figure 2). *R*_y_, *R*_x_, and *R*_b,*i*_, denote isotope ratios measured in the respective spike, sample, and *i*th blend solution related to the isotope of major abundance (reference isotope; in the case of enriched silicon: ^28^Si; *R*_x_ = *I*(^30^Si)/*I*(^28^Si), with the intensities *I* of the measured isotope signals); *M*_x_ and *M*_y_ are the molar masses of the respective analyte and spike material, which need not be known. The only requirement is the knowledge of mass fraction *w*_z_ of the element of interest in reference material z. This is given in a calibration certificate or accessible via a separate measurement (gravimetric preparation).

One advantage of this new method is based on the fact that the measured intensity ratios are entered into Equation (1) directly without the need for mass bias correction. Usually, in ICP–MS, measured isotope ratios suffer from mass discrimination effects (mass bias) and have to be corrected by the use of calibration *K* factors. The latter are accessible via reference materials, applications of empirical relations like the exponential law, or gravimetrical methods [3,5,28]. In the set of equations used in this work, however, the *K* factors applied to measured isotope ratios cancel out, enabling the direct use of the measured ratios and, thus, simplifying this method for convenience. The only exception is spike ratio *R*_y_, for which no value measured in the matrix is available. Reasonable spike materials feature *R*_y_ > 100. In this case, mixing ratio *R*_y_, measured in a matrix different from that of the sample matrix, with *R*_x_ and *R*_b,*i*_ measured in the sample matrix, changes the result (*w*_x_) well within its uncertainty and, therefore, is insignificant.

As stated above, Equation (9) in [1] has been analytically derived again in this work from scratch. This derivation is given in the Appendix A. Equation (3) displays the analytical solution for mass fraction *w*_x_ in the notation used in this work. This equation is exactly the same as Equation (9) in [1]. Additionally, mass fraction *w*_z_ of the reference material used has to be known. Masses (*m_j_*_,*i*_) of components *j* (*j* = sample x, reference z, or spike y) in blends *i* (*i* = 1, 2, 3) have to be known (gravimetric preparation) as well as the respective isotope ratios, namely, the *R*_b3_ reads:isotope ratio (monitor vs. reference isotope) in Blend 3, the *R*_z,2_ reads:isotope ratio (monitor vs. reference isotope) in reference z, and *x*_z,1_ reads:amount-of-substance fraction *x* of Isotope 1 (reference isotope) in reference material z. Note that the “reference” isotope is the isotope of the highest abundance in the sample, whereas “reference” material is a characterized material with (almost) the same isotopic composition as the sample. In the case of different isotopic compositions of x and z, the molar masses, *M*_x_ (sample) and *M*_z_ (reference), and the amount-of-substance fractions, *x*_z,1_ and *x*_x,1_, have to be known. When applying Equation (3), three gravimetrically prepared blends are required.
(3)$$w_{x}=w_{z}\left[\frac{\begin{array}{cc} \; & m_{y1}m_{z2}m_{y3}\left(R_{b2}-R_{z,2}\right)\left(R_{b3}-R_{b1}\right)\\ \; & -m_{y1}m_{y2}m_{z3}\left(R_{b3}-R_{z,2}\right)\left(R_{b2}-R_{b1}\right)\\ \; & -m_{z1}m_{y2}m_{y3}\left(R_{b1}-R_{z,2}\right)\left(R_{b3}-R_{b2}\right) \end{array}}{\begin{array}{cc} \; & -m_{y1}m_{x2}m_{y3}\left(R_{b2}-R_{x,2}\right)\left(R_{b3}-R_{b1}\right)\\ \; & +m_{y1}m_{y2}m_{x3}\left(R_{b3}-R_{x,2}\right)\left(R_{b2}-R_{b1}\right)\\ \; & +m_{x1}m_{y2}m_{y3}\left(R_{b1}-R_{x,2}\right)\left(R_{b3}-R_{b2}\right) \end{array}}\right]\frac{x_{z,1}M_{x}}{x_{x,1}M_{z}}$$

## 3. Materials and Experimental Methods

In the first test system, the sample solutions, x, consisted of aqueous TMAH (*w*(TMAH) = 0.0006 g/g) as the matrix with traces of the analyte: silicon with natural isotopic composition. The sample solutions were taken from a study described in [29]. There, the aqueous TMAH served as a blank solution for the measurement of isotope ratios of enriched silicon samples. Briefly, the TMAH sample solutions were prepared by dissolving highly concentrated (*w*(TMAH) = 0.25 g/g), electronic grade (99.9999%) TMAH from Alfa Aesar™ (Thermo Fisher (Kandel) GmbH, Germany) in highly purified water (resistivity = 18 MΩ cm), the latter prepared using a commercial water purification system (Merck Millipore™, Burlington, MA, USA). All solutions were prepared and stored in precleaned labware (perfluoroalkoxy copolymer PFA) according to a protocol given in [29]. The spike solutions, y, originated from chemically pure silicon single crystals highly enriched in ^30^Si dissolved in aqueous TMAH, whereas reference material z was prepared from well-characterized silicon crystals (material code: WASO04) with known natural-like isotopic composition [30]. All stock solutions, dilutions, and blends in this study were prepared gravimetrically, applying air buoyancy correction [31]. The silicon crystals used for the stock solutions of y (spike) and z (reference) were cleaned and etched prior to dissolution with the final mass fractions prior to blend preparation: *w*_z_ = 4 µg/g; *w*_y_ = 2 µg/g. The isotopic composition of y as well as the respective “true” isotope ratios, *R*_y,_ were taken from [20]. For the determination of *w*_x_ according to Equations (1) and (2), a series of five blends, b*_i_*, was prepared, each consisting of approximately *m*_x,*i*_ = 10 g, *m*_y,*i*_ = 22.5 g, and stepwise increasing amounts of solution z with *m*_z,1_ = 0 g, *m*_z,2_ = 7.5 g, *m*_z,3_ = 10 g, *m*_z,4_ = 14.5 g, and *m*_z,5_ = 23 g (compare Figure 1).

The sample preparation for sulfur determination in the biodiesel fuel matrix is described in [23]. Briefly, an ampoule (15 mL) from the CCQM-K123 study containing commercial biodiesel fuel doped with the target analyte sulfur using dibutyl sulfide was used as the sample. Defined aliquots of reference z (NIST SRM 3154) and spike y (BAM S-34) material were added prior to the digestion. For the blend solutions, acid digestion was applied (HNO_3_/H_2_O_2_) using a high-pressure asher system (Anton Paar, Graz, Austria), with *ϑ*_max_ ≈ 300 °C and *p*_max_ ≈ 130 bar. After digestion, the solution was evaporated to dryness, redissolved with 2 mL HNO_3_ (0.028 mol/L), and loaded on chromatographic columns filled with 1 mL AG 1X8 resin. The sample matrix was eluted with water, and, subsequently, the sulfur was eluted with 8 mL HNO_3_ (0.25 mol/L). Samples were evaporated to dryness and redissolved in HNO_3_ (0.02 kg/kg) to adjust a sulfur mass fraction of 2 mg/kg. Prior to the MC–ICP–MS measurements, sodium was added so that a final Na mass fraction of 4 mg/kg was achieved. The addition of sodium significantly increases sensitivity and prevents losses of sulfur through the membrane of the desolvating nebulizer system, presumably as sulfuric acid [32]. The complete analytical procedure was validated by applying the double IDMS calibration approach with reference material NIST SRM 2723a. The determined sulfur mass fraction was (10.94 ± 0.10) mg/kg, while the certified sulfur mass fraction was (10.90 ± 0.31) mg/kg, with *k* = 2 in both cases.

For the measurement of the third system, TRF in human serum, reference material ERM^®^-DA470k/IFCC “Human Serum” (Merck KGaA, Darmstadt, Germany) was applied as the reference, and Seronorm^TM^ Immunoprotein Lyo L-1 (Invicon GmbH, Munich, Germany), which has certified values and ranges in accordance with the Guidelines of the German Medical Association (Rili-BÄK) of 2014, was used as the sample [33]. Both materials were reconstituted according to the manufacturers’ protocols and thereafter used for sample preparation [33]. The iron saturation procedure was based on the method described by del Castillo Busto et al. and C. Frank et al. except for some optimizations [15,34]. For the individual blends, a solution containing 7.5 to 22.5 mg of an iron solution (250 µg/g in 2.5% HNO_3_, dilution from BAM A-primary-Fe-2; BAM, Berlin, Germany), 5 to 10 mg sodium carbonate (Na_2_CO_3_) solution (500 mmol/kg; BioUltra, Merck KGaA, Darmstadt, Germany), and 50 to 277 mg tris(hydroxymethyl)methylamine (Tris) buffer solution (12.5 mmol/kg, adjusted to *p*H 6.4 with acetic acid; BioUltra, Merck KGaA, Darmstadt, Germany) was mixed, and 0 to 200 mg of the reconstituted ERM^®^-DA470k/IFCC serum and 50 mg of the reconstituted Seronorm^TM^ Immunoprotein Lyo L-1 serum were added. The amount of iron and Na_2_CO_3_ solution was adjusted to the amount of added human serum, which increased from Blend 1 to Blend 4. All solutions were incubated at room temperature for 1.5 h. Then, 160 mg of the TRF spike was added. In Section 4.3, the added amounts of reference, sample, and spike solution for the different blends are given. 

The TRF spike was prepared in advance from human apo-TRF purchased from Sigma-Aldrich (St Louis, MO, USA) and ^54^Fe spike from Trace Science International Corp. (Ontario, ON, Canada), with a certified isotopic abundance of ^54^Fe 99.86%, ^56^Fe 0.11%, ^57^Fe 0.01%, and ^58^Fe 0.02%. For this, 20 mg of apo TRF was dissolved in a solution containing 0.435 g Na_2_CO_3_ (500 mmol/kg), 4.3 g Tris (12.5 mmol/kg), and 0.4 g ^54^Fe (250 µg/g), adjusted to *p*H = 8 with 0.15 mmol/kg HNO_3_. After an incubation time of 3 h at room temperature, the solution was purified with a PD-10 desalting column (Cytiva GmbH, Freiburg im Breisgau, Germany), according to the manufacturer’s protocol. The spike solution was freeze-dried, and the solid was used for the preparation of a TRF spike solution of 2.3 mg/g, which was used to prepare the blends.

After the gravimetric sample and blend preparation, in all blends, lipoprotein precipitation was performed by adding 5 µL of a solution containing 200 mg magnesium chloride (Merck KGaA, Darmstadt, Germany) and 100 mg dextran sulfate sodium salt (Merck KGaA, Darmstadt, Germany) in 1 mL ultrapure water, incubating for 30 min at 4 °C and centrifuging the samples at 12,000× *g* for 10 min. Each supernatant was purified with PD MidiTrap columns (Cytiva GmbH, Freiburg im Breisgau, Germany), according to the manufacturer’s protocol. The solutions obtained were used for the HPLC/ICP–MS analysis.

All three sample/matrix systems were investigated using ICP–MS instrumentation. The respective operation parameters of the isotope ratio measurements are summarized in Table 1. As an example, the Si in TMAH measurements were run in six sequences measuring the intensities of all three stable Si isotopes (^28^Si, ^29^Si, and ^30^Si) in each blend, b*_i_* (*i* = 1…5).

## 4. Results and Discussion

### 4.1. Silicon in Aqueous TMAH

The first example of the application of the new combined IDMS-Standard Addition approach is the determination of silicon with natural isotopic composition in the matrix of aqueous TMAH. As an example, the essential input and output quantities of one experimental run (Sequence 1) are given in Table 2. The corresponding linear regression curve is displayed in Figure 3.

As a direct consequence of linear regression analysis, mass fraction *w*_x_ of the analyte in the respective sample x (TMAH_aq_) was derived according to Equation (2). A set of 5 blends is sufficient to obtain proper regression statistics.

The individual results of *w*_x_(Si), obtained from six sequences (runs), are displayed in Figure 4 (left) and Table 3. They yield an average of *w*_x_(Si) = 0.081(73) µg/g. As proof of consistency of the individual measurement results, the concept of degrees of equivalence (*d*_i_) was applied. The individual *d_i_* of the respective measurements are plotted in Figure 4 (right). All data encompass the zero line with their associated expanded uncertainties, which is an approval of the complete consistency of the data.

### 4.2. Sulfur in Biodiesel Fuel (BDF)

The determination of sulfur in BDF, according to Equations (1) and (2), was performed in a similar way using the five blends, *b_i_*. The essential input and output quantities of one experimental run are given in Table 4. The corresponding linear regression curve of the respective dataset is displayed in Figure 5.

Linear regression analysis yields the mass fraction (*w*_x_) of the analyte (sulfur) in the respective sample x (BDF), according to Equation (2).

The determination of sulfur in BDF, according to Equations (1) and (2), yields an average mass fraction of *w*_x,corr_(S) = 7.36(11) µg/g. The individual results of *w*_x,corr_(S), measured in three runs, are shown in Figure 6 (left) and Table 5. Additionally, as proof of consistency, the concept of degrees of equivalence, *d_i_*, was applied. The *d_i_* of the respective measurements are plotted in Figure 6 (right). The single results encompass the zero line with their associated expanded uncertainties, which proves the complete consistency of the data. In the case of sulfur determination in this work, the respective sample was taken from a stock solution already used in the interlaboratory comparison CCQM-K123, as described in [23]. There, the sulfur determination was carried out using double IDMS, yielding *w*_x_ = 7.394(46) µg/g. The corresponding result is shown in Figure 6 (left, red solid line) for comparison. Both the IDMS and the new combined IDMS/standard addition approach show excellent agreement within the limits of uncertainty.

### 4.3. Transferrin in Human Serum

The determination of TRF in human serum, according to Equations (1) and (2), was performed in a similar way using four blends, *b_i_*. The essential input and output quantities of one experimental run are given in Table 6. The corresponding linear regression curve of the respective dataset is displayed in Figure 7.

The linear regression analysis yields mass fraction *w*_x_ of the analyte TRF in the respective sample x (human serum) according to Equation (2).

The individual results of *w*_x_(TRF), obtained from two independent measurement sequences, are displayed in Figure 8 (left). Both values, *w*_x_(TRF) = 2.128(45) mg/g and *w*_x_(TRF) = 2.340(76) mg/g for *k* = 1, are consistent with the certified target value and its allowed range [33]. As proof of consistency of the individual measurement results, the concept of degrees of equivalence, *d*_i_, was applied. The individual *d_i_* of the respective measurements are plotted in Figure 8 (right). All data encompass the zero line with their associated expanded uncertainties.

### 4.4. Comparison of Linear Regression (This Work) and Analytical Solution (Pagliano and Meija)

The comparison of the two approaches, combining IDMS and standard addition presented in this work and in [1], was performed using the dataset of silicon determination in the TMAH matrix (five available blends: b_1_, b_2_, b_3_, b_4_, b_5_). Figure 9 shows the results of *w*_x_(Si), determined in six sequences (runs), using five blends (this work) and three blends (b_2_, b_3_, b_4_; approach of [1] using Equation (3)). At first glance, the associated uncertainties of the results determined in this work (black circles) are significantly smaller than the respective uncertainties of the results determined using the approach of [1] (red circles).

Prior to this comparison, both approaches (this work and that of [1]) were checked for consistency using simulated ideal data: both methods yielded exactly the same result. However, when using real experimental data (e.g., the results of the five blends for silicon determination), the results of the approach of [1] depend strongly on the input data (the three blends that were used for analysis). Table 7 shows the different numerical results between the two approaches of a representative run. In the case of this work, all five blends were used (left column), whereas, for the approach of [1], triple blends b_1_, b_2_, b_3_ or b_2_, b_3_, b_4_, or b_3_, b_4_, b_5_ each yielded extremely different results. The analytical approach (Equation (9) of [1], which is equal to Equation (3) in this publication) is based on a set of three blends only, without the need for the measurement of *R*_y_, which appears initially as a benefit. However, the analysis shows that the three-blend application leads, in some cases, to unrealistic (not accurate) results (results of triples b_1_, b_2_, b_3_ and b_3_, b_4_, b_5_). Equation (9) in [1] reacts extremely sensitively towards that kind of input data. In our case, the three inner blends yield the most reasonable (“accurate”) result, e.g., the first blend, b_1_, includes no reference material z (*m*_z_ = 0 g). This suggests a rather unstable and sensitive applicability of Equation (9) of [1] compared to the linear regression approach of this work. Moreover, for practical applications, Equations (1) and (2) are much simpler and user-friendly, and the respective evaluations and measurements are much easier than in the case of the approach used in [1], although the latter is correct from the mathematical point of view.

For the comparison of the uncertainties associated with *w*_x_, which is the second-most important outcome of the respective approach, we chose the results of the blend triple b_2_, b_3_, b_4_ of the six sequences. As can be seen in Figure 9, the uncertainties obtained using the approach of this work are significantly smaller than those obtained using Equation (9) from [1].

An uncertainty analysis (Table 8) of Equation (3) (same as Equation (9) in [1]) according to GUM [9], using GUM Workbench Pro™ software (version 2.4.1 392, Metrodata GmbH, Germany), shows that the absolute values of the sensitivity coefficients of the main uncertainty contributions, *R*_b2_, *R*_b1_, and *R*_b3_, range in the 10^1^-10^2^ region. This seems rather high and is mainly responsible for the elevated uncertainty compared to the smaller uncertainty obtained using linear regression in the approach of this work.

## 5. Conclusions

The new method presented—a combination of IDMS and standard addition—for absolute elemental quantification is recommended for sample analytes with two or more isotopes, especially when appearing in traces (ng/g) and/or in a complex matrix that is difficult or (almost) impossible to separate. The three different examples of this study demonstrate the convenient applicability of this approach successfully. Moreover, the second example (determination of sulfur in biodiesel fuel) acts as a validation system since the same sample material has been measured previously and independently using a double IDMS method in the context of an international key comparison (CCQM-K123), yielding excellently matching results within the limits of uncertainty. Mainly, since the blends are measured, all the “measurement” solutions are matrix-matched; thus, potential matrix effects are also canceled out. The comparison of the new method (linear regression) with the analytical approach reported in [1] shows the formal (mathematical) agreement of both methods. However, the use of experimental data suggests a more robust behavior of the approach presented in this work, yielding more “accurate” results with significantly lower associated uncertainties (see calculation examples in the Appendix A). Moreover, the new approach is more practical in daily laboratory work due to a simpler evaluation algorithm that might be less fault-prone.

## Figures and Tables

**Figure 1 molecules-26-02649-f001:**
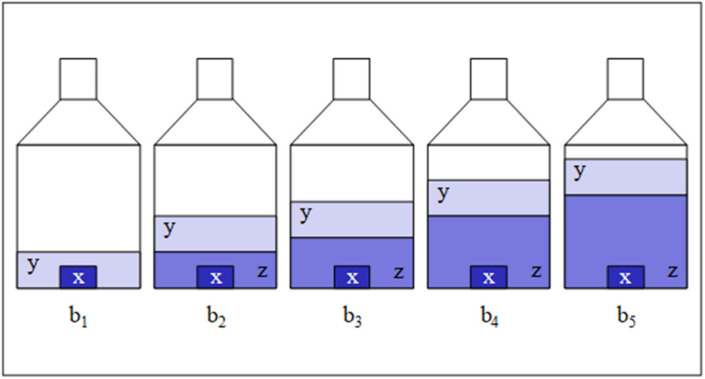
Schematic of a set of 5 blends (b*_i_*) gravimetrically prepared from approximately the same masses (*m*_x,*i*_) of analyte sample x, same masses (*m*_y,*i*_) of spike solution y, and different masses (*m*_z,*i*_) of reference solution z (*m*_z,1_ < *m*_z,2_ < *m*_z,3_ < *m*_z,4_ < *m*_z,5_; in this work, *m*_z,1_ = 0 g).

**Figure 2 molecules-26-02649-f002:**
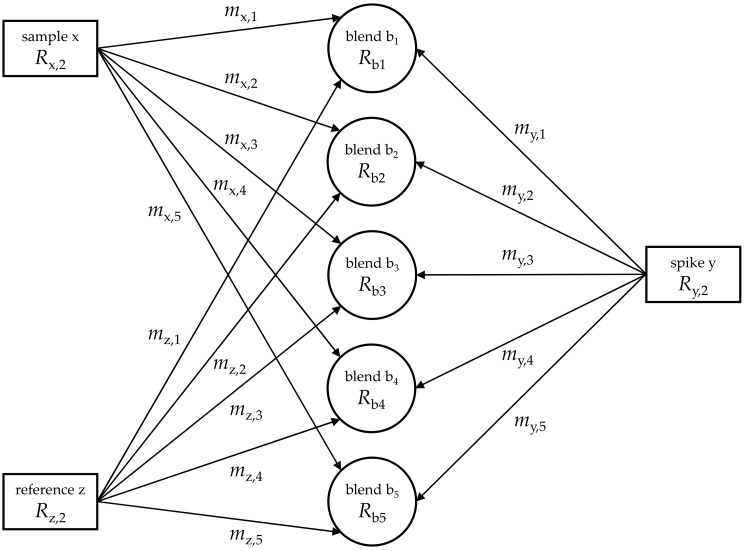
Relationship of the initial components (sample x, spike y, and reference z) and the respective blends (b*_i_*), indicating (a) the abundances in which the components were blended and (b) the quantities, which have to be measured to be able to calculate analyte mass fraction *w*_x_ in the sample.

**Figure 3 molecules-26-02649-f003:**
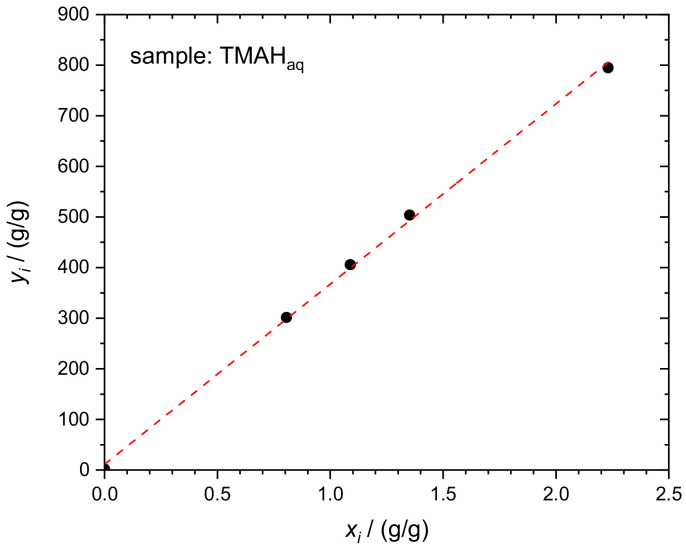
Linear regression evaluation of the Si mass fraction in TMAH_aq_ according to Equation (2). Dataset in Table 2.

**Figure 4 molecules-26-02649-f004:**
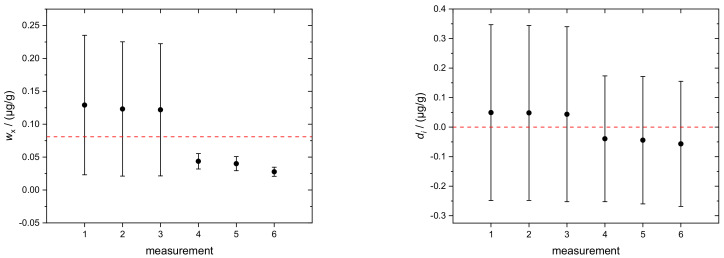
(**Left**): Mass fraction *w*_x_ (Si) in TMAH_aq_. Error bars denote combined uncertainties (*k* = 1). The red dashed line indicates the average value. (**Right**): Degrees of equivalence *d_i_* of the respective measurement results. Error bars indicate expanded uncertainties (*k* = 2) associated with *d_i_*. All single results are consistent with the average value since the respective uncertainties encompass the red dashed zero line.

**Figure 5 molecules-26-02649-f005:**
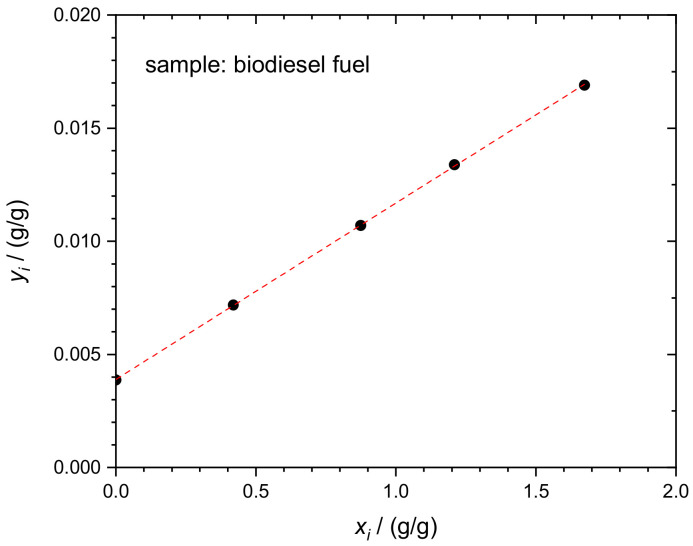
Linear regression of the S mass fraction in biodiesel fuel according to Equation (2). Dataset in Table 4.

**Figure 6 molecules-26-02649-f006:**
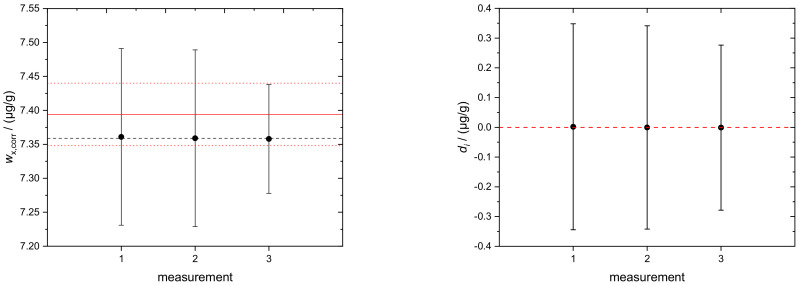
(**Left**) Black circles: corrected mass fractions *w*_x,corr_(S) of sulfur in BDF and the associated uncertainties (*k* = 1); black dashed line: average of three single runs. Data from the new combined IDMS/standard addition approach (this work). Red solid line: average value of *w*(S) from BAM, applying IDMS and using the same solution [23]. Upper and lower associated uncertainties: red dotted lines. (**Right**) Degrees of equivalence (*d_i_*) of the respective measurement results. Error bars indicate expanded uncertainties (*k* = 2) associated with *d_i_*. All single results are consistent with the average value since the respective uncertainties encompass the red dashed zero line.

**Figure 7 molecules-26-02649-f007:**
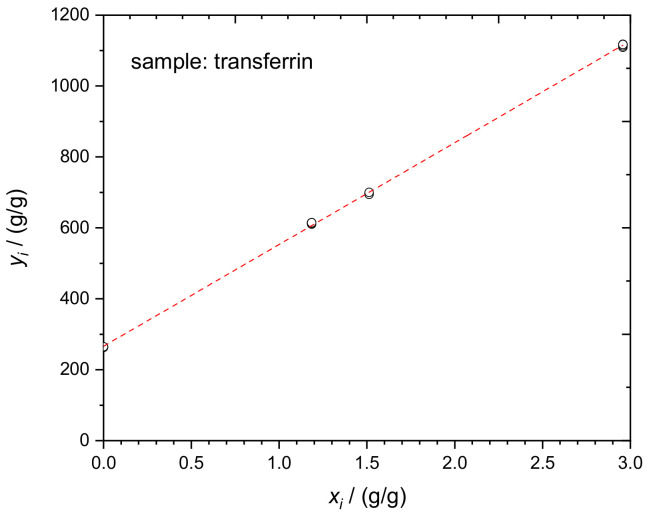
Linear regression evaluation of the TRF mass fraction in human serum according to Equation (2). Dataset in Table 6.

**Figure 8 molecules-26-02649-f008:**
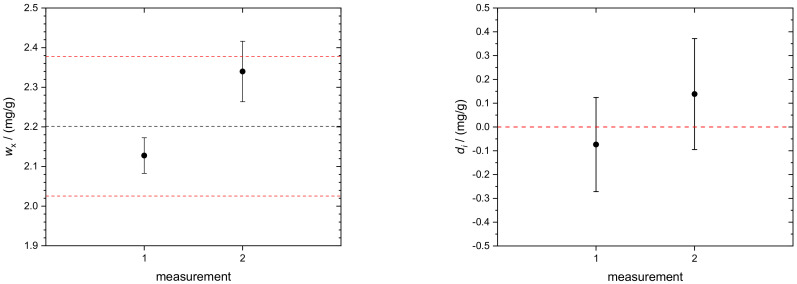
(**Left**) Black circles: mass fractions *w*_x_(TRF) of transferrin in human serum and the associated uncertainties (*k* = 1); dashed black line: certified value of the Seronorm sample (for instrument, see Beckmann, AU). Data from the new combined IDMS/standard addition approach (this work). Red dashed lines: allowed target range according to Rili-BÄK [33]. (**Right**) Degrees of equivalence, *d_i_*, of the respective measurement results. Error bars indicate expanded uncertainties (*k* = 2) associated with *d_i_*. All single results are consistent with the certified value since the respective uncertainties encompass the red dashed zero line.

**Figure 9 molecules-26-02649-f009:**
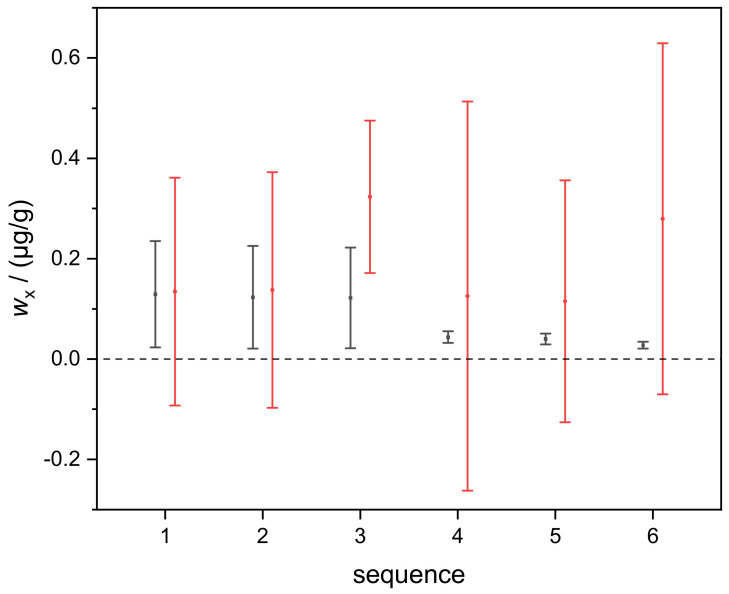
Black circles (error bars): mass fractions *w*_x_(Si) and the associated uncertainties (*k* = 1) of six runs determined using the approach of this work. Red circles (error bars): results of the same input data (blends b_2_, b_3_, b_4_ only) using the approach of [1].

**Table 1 molecules-26-02649-t001:** Operation parameters of the mass spectrometric isotope ratio measurements applied for the three sample/matrix systems. (SP = spike; *K* = *K* factor).

Sample/Matrix	Silicon/TMAH	Sulfur/Biodiesel Fuel	TRF/Human Serum
Laboratory	PTB	BAM	PTB
Instrument	Thermo MC–ICP–MS Neptune	Thermo MC-ICP-MS Neptune Plus	Agilent 8900 ICP-QQQ-MS
Sample Introduction	PFA nebulizer 100 µL/min PEEK/PFA cyclonic + Scott chamber sapphire torch + injector BN shield Ni sampler + Ni X-skimmer	Aridus II desolvating system PFA nebulizer 100 µL/min Aridus PFA spray chamber standard torch and injector quartz shieldNi sampler + Ni H-skimmer	PFA MicroFlow nebulizer 700 µL, Scott chamber at 3 °C torch with 1 mm injector Pt shield Pt sampler and skimmer
Gas Flow Rates (Ar)	cooling: 16 L min^−1^ auxiliary: 0.8 L min^−1^ sample: 1.0 L min^−1^	cooling: 16 L min^−1^ auxiliary: 0.9 L min^−1^ sample: 0.85 L min^−1^	cooling: 15 L min^−1^ auxiliary: 0.9 L min^−1^ nebulizer gas: 0.8 L min^−^^1^ reaction gas (H_2_): 6.1 mL min^−^^1^
Machine Parameters	high resolution (*M*/∆*M* = 8000) RF power 1180 W integration time 4.194 s idle time 3 s number of blocks 6 cycles/block 3 rotating amplifiers: yes Faraday cups: L3(^28^Si), C(^29^Si), H3(^30^Si)	high resolution (*M*/∆*M* = 8000) RF power 1200 W integration time 4.194 s idle time 3 s number of blocks 1 cycles/block 40 rotating amplifiers: no Faraday cups: L3(^32^S), C(^33^S), H3(^34^S)	MS/MS mode RF power 1550 W Sample depth 8.0 mm x-lens configuration integration time 0.1 s *m*/*z* 53, 54, 56, 57, 58, 60
Sequence Settings	rinse time 120 s take-up time 60 s measured samples/sequence b1, b2, b3, b4, b5 (4 times each)	rinse time 30 s take-up time 80 s measured samples/sequence b1, b2, b3, b4, b5 (3 times each, separated by a block of 5 standards)	rinse time + take-up not applicable: HPLC separationmeasured samples/sequence b1, b2, b3, b4, blank, SP, *K*, blank (4 times)
Separation Settings			Agilent Bioinert 1260 HPLC system Column: MonoQ^®^ GL 5/50 from GE Healthcare (Uppsala, Sweden) Mobile phase A: 12.5 mmol/L Tris at *p*H = 6.4 Mobile phase B: 12.5 mmol/L Tris + 125 mmol/L NH_4_Ac at *p*H = 6.4 Flow: 0.5 mL min^−1^ Gradient: 0 min → 0% B, 20 min → 100% B, 27 min → 100% B Column oven 30 °C Injection volume: 10 µL MWD 254 nm, 280 nm

**Table 2 molecules-26-02649-t002:** Determination of *w*_x_(Si) in TMAH_aq_. The relevant input and output data of the linear regression analysis are given for a single representative dataset (Sequence 1), with *w*_z_ = 4.0069 µg/g.

	x	z	y					
b*_i_*	TMAH_aq_	WASO04	“Si30”	*R* _b*,i*_			*R* _x*,i*_	*R* _y*,i*_
*i*	*m* _x,*i*_	*m* _z,*i*_	*m* _y,*i*_	*I*(^30^Si)/*I*(^28^Si)	*x_i_*	y*_i_*	*I*(^30^Si)/*I*(^28^Si)	*I*(^30^Si)/*I*(^28^Si)
	g	g	g	V/V	g/g	g/g	V/V	mol/mol
1	10.0863	0.0000	22.8557	113.77732	0.0000	1.80	0.03353	204.19578
2	9.7836	7.8858	22.7577	1.59655	0.8060	301.51	0.03353	204.19578
3	9.6700	10.5255	22.3198	1.18847	1.0885	405.71	0.03353	204.19578
4	11.3192	15.3000	22.4864	0.83531	1.3517	503.86	0.03353	204.19578
5	10.0440	22.4061	22.7651	0.61405	2.2308	794.84	0.03353	204.19578
					*a* _1_	*a* _0_	*w* _x_	
					(g/g)/(g/g)	(g/g)	µg/g	
					356.10062	11.474	0.13	

**Table 3 molecules-26-02649-t003:** Mass fractions *w*_x_ (Si) of silicon in TMAH_aq_ and the associated uncertainties (*k* = 1).

Run	*w*_x_ (Si)	*u*(*w*_x_ (Si))
	µg/g	µg/g
1	0.13	0.11
2	0.12	0.10
3	0.12	0.10
4	0.044	0.012
5	0.040	0.011
6	0.028	0.007
average	0.081	0.073

**Table 4 molecules-26-02649-t004:** Determination of *w*_x_ in BDF. The relevant input and output data of linear regression analysis are given for a single representative dataset (1st run), with *w*_z_ = 16.145 µg/g. A procedural blank was subtracted from the result. The procedural blank was determined during an external measurement, as described in [23].

	x	z	y					
b*_i_*	BDF	NIST SRM 3154	BAM S-34	*R* _b*,i*_			*R* _x*,i*_	*R* _y*,i*_
*i*	*m* _x,*i*_	*m* _z,*i*_	*m* _y,*i*_	*I*(^32^S)/*I*(^34^S)	*x_i_*	*y_i_*	*I*(^32^S)/*I*(^34^S)	*I*(^32^S)/*I*(^34^S)
	g	g	g	V/V	g/g	g/g	V/V	mol/mol
1	0.23748	0.00000	0.09670	0.20030	0.00000	0.00387	21.16643	0.00099
2	0.24149	0.10125	0.09843	0.36731	0.41927	0.00718	21.16643	0.00099
3	0.23819	0.20828	0.10899	0.48453	0.87443	0.01070	21.16643	0.00099
4	0.25126	0.30375	0.10631	0.64994	1.20891	0.01339	21.16643	0.00099
5	0.24311	0.40679	0.09966	0.83887	1.67328	0.01690	21.16643	0.00099
					*a* _1_	*a* _0_	*w* _x_	*w* _x,corr_
					(g/g)/(g/g)	g/g	µg/g	µg/g
					0.007797	0.003894	8.063	7.36

**Table 5 molecules-26-02649-t005:** Blank corrected mass fractions *w*_x,corr_(S) of sulfur in biodiesel fuel (uncertainties with *k* = 1).

Run	*w*_x,corr_(S)	*u*(*w*_x,corr_(S))
	µg/g	µg/g
1	7.36	0.13
2	7.36	0.13
3	7.358	0.079
average	7.36	0.11

**Table 6 molecules-26-02649-t006:** Determination of *w*_x_(TRF) in human serum. The relevant input and output data of linear regression analysis are given for a single representative dataset (M21-3), with *w*_z_ = 2.296 mg/g.

	x	z	y					
b*_i_*	Seronorm^TM^ Immuno-Protein Lyo L-1	ERM^®^-DA470k/IFCC	In-House Prepared TRF Spike	*R* _b*,i*_			*R* _x*,i*_	*R* _y*,i*_
*i*	*m* _x,*i*_	*m* _z,*i*_	*m* _y,*i*_	*R*(^54^Fe/^56^Fe)	*x_i_*	*y_i_*	*R*(^54^Fe/^56^Fe)	*R*(^54^Fe/^56^Fe)
	g	g	g	mol/mol	g/g	g/g	mol/mol	mol/mol
1	0.04848	0.00000	0.15104	3.01104	0.00000	262.4	0.063703	251.22
2				2.99958	0.00000	263.4		
3				2.99154	0.00000	264.1		
4				2.98887	0.00000	264.4		
5	0.04923	0.05834	0.15123	1.31282	1.18510	614.6		
6				1.32126	1.18510	610.5		
7				1.31656	1.18510	612.8		
8				1.31365	1.18510	614.2		
9	0.04910	0.07424	0.14909	1.15279	1.51220	697.3		
10				1.15426	1.51220	696.3		
11				1.15733	1.51220	694.4		
12				1.14892	1.51220	699.8		
13	0.04923	0.14561	0.14951	0.74959	2.95788	1109.1		
14				0.74722	2.95788	1113.0		
15				0.74440	2.95788	1117.6		
16				0.74494	2.95788	1116.7		
					*a* _1_	*a* _0_	*w* _x_	
					(g/g)/(g/g)	g/g	mg/g	
					287.078	266.04	2.128	

**Table 7 molecules-26-02649-t007:** A representative result of mass fractions *w*_x_(Si) determined using the approach of this work (using all five blends) and the approach of [1] (using the three triple blends: b_1_, b_2_, b_3_; b_2_, b_3_, b_4_; and b_3_, b_4_, b_5_). Only the combination of b_2_, b_3_, b_4_ (the inner blends) yielded a reasonable numerical result.

This Work	Approach of [1]
blends	blends
b_1_, b_2_, b_3_, b_4_, b_5_	b_1_, b_2_, b_3_
*w*_x_(Si)	*w*_x_(Si)
µg/g	µg/g
0.1292	−0.5004

**Table 8 molecules-26-02649-t008:** A representative uncertainty budget of mass fraction *w*_x_(Si) determined using the approach of [1] (using the blends b_2_, b_3_, b_4_). The main contributions originate from *R*_b2_, *R*_b1_, and *R*_b3_, with the largest absolute values of respective sensitivity coefficients.

Quantity	Unit	Best Estimate (Value)	Standard Uncertainty	Sensitivity Coefficient	Index
*X_i_*	[*X_i_*]	*x_i_*	*u*(*x_i_*)	*c_i_*	
*w* _z_	µg/g	4.00694	6.01 × 10^−3^	0.031	0.0%
*m* _y1_	g	22.49660	1.00 × 10^−3^	2.2	0.0%
*m* _z2_	g	10.31270	1.00 × 10^−3^	11	0.0%
*m* _y3_	g	22.26850	1.00 × 10^−3^	2.8	0.0%
*R* _b2_	V/V	1.23933	3.62 × 10^−3^	92	74.4%
*R* _z2_	V/V	0.033527	335 × 10^−6^		
*R* _b3_	V/V	0.88472	1.50 × 10^−3^	−73	8.0%
*R* _b1_	V/V	1.68846	5.46 × 10^−3^	−30	17.5%
*m* _y2_	g	22.43470	1.00 × 10^−3^	−5.0	0.0%
*m* _z3_	g	14.57170	1.00 × 10^−3^	−4.2	0.0%
*m* _z1_	g	7.46840	1.00 × 10^−3^	−6.4	0.0%
*m* _x2_	g	9.22730	1.00 × 10^−3^	0.33	0.0%
*R* _x2_	V/V	0.033527	335 × 10^−6^	10	0.0%
*m* _x3_	g	9.79000	1.00 × 10^−3^	−0.13	0.0%
*m* _x1_	g	9.46490	1.00 × 10^−3^	−0.20	0.0%
***w*_x_**	**g**	**0.125**	**0.388**		

## Data Availability

The data that support the findings of this study are available from the authors upon request.

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
