# Peer review of "Combining Isotope Dilution and Standard Addition—Elemental Analysis in Complex Samples"

_molecules, 2021, doi:10.3390/molecules26092649_

Round 1

Reviewer 1 Report

This manuscript reported a new method combining isotope dilution mass spectrometry (IDMS) and standard addition to determine the mass fractions of different elements in challenging matrices. It is technically sound. However, the presentation could be improved,

  1. The manuscript has lots of long sentences, for example, the first sentence of introduction has four lines. Rewrite them to make it easier to understand.
  2. Line 115, the author mentioned that this new approach of combining IDMS with standard addition is for the first time used in a complex biological sample, actually, similar method has been reported, although the purpose is different. Please see Larios, R. et al. Accurate quantification of carboplatin adducts with serum proteins by monolithic chromatography coupled to ICPMS with isotope dilution analysis. J. Anal. At. Spectrom. 2019, 34, 729-740.
  3. Table 1, the first letter of the words should be capitalized.
  4. The title used challenging, and it could be replaced by complicated or more objective expressions. Or else define challenging.  

Reviewer 2 Report

Dear Authors,

Identification of low concentrated molecules in complex matrix samples is a very challenging task, therefore the presented topic is noteworthy. I can imagine that you were searching for suitable examples of samples characterized by complex matrix, however for the identification of single proteins in serum samples (case of TRF), well defined antibody-based arrays are already used for decades. Therefore, it would be appreciated to mention alternative methods of studied molecules identification. Also, the first part of introduction needs grammatical improvement.

Sincerely,
